# Long-Term Seizure Outcomes and Predictors in Patients with Dysembryoplastic Neuroepithelial Tumors Associated with Epilepsy

**DOI:** 10.3390/brainsci13010024

**Published:** 2022-12-22

**Authors:** Huawei Zhang, Yue Hu, Adilijiang Aihemaitiniyazi, Tiemin Li, Jian Zhou, Yuguang Guan, Xueling Qi, Xufei Zhang, Mengyang Wang, Changqing Liu, Guoming Luan

**Affiliations:** 1Department of Neurosurgery, Sanbo Brain Hospital, Capital Medical University, Beijing 100093, China; 2Department of Neurosurgery, Aviation General Hospital, China Medical University, Beijing 100012, China; 3Beijing Institute of Brain Disorders, Capital Medical University, Beijing 100093, China; 4Department of Pathology, Sanbo Brain Hospital, Capital Medical University, Beijing 100093, China; 5Department of Radiology, Sanbo Brain Hospital, Capital Medical University, Beijing 100093, China; 6Department of Neurology, Sanbo Brain Hospital, Capital Medical University, Beijing 100093, China

**Keywords:** dysembryoplastic neuroepithelial tumors, epilepsy, surgery, seizure outcome, prognosis

## Abstract

Objective: To determine the predictors and the long-term outcomes of patients with seizures following surgery for dysembryoplastic neuroepithelial tumors (DNTs); Methods: Clinical data were collected from medical records of consecutive patients of the Department of Neurosurgery of Sanbo Brain Hospital of Capital Medical University with a pathological diagnosis of DNT and who underwent surgery from January 2008 to July 2021. All patients were followed up after surgery for at least one year. We estimated the cumulative rate of seizure recurrence-free and generated survival curves. A log-rank (Mantel–Cox) test and a Cox proportional hazard model were performed for univariate and multivariate analysis to analyze influential predictors; Results: 63 patients (33 males and 30 females) were included in this study. At the final follow-up, 49 patients (77.8%) were seizure-free. The cumulative rate of seizure recurrence-free was 82.5% (95% confidence interval (CI) 71.8–91.3%), 79.0% (95% CI 67.8–88.6%) and 76.5% (95% CI 64.8–87.0%) at 2, 5, and 10 years, respectively. The mean time for seizure recurrence-free was 6.892 ± 0.501 years (95% CI 5.91–7.87). Gross total removal of the tumor and a short epilepsy duration were significant predictors of seizure freedom. Younger age of seizure onset, bilateral interictal epileptiform discharges, and MRI type 3 tumors were risk factors for poor prognosis; Conclusions: A favorable long-term seizure outcome was observed for patients with DNT after surgical resection. Predictor analysis could effectively guide the clinical work and evaluate the prognosis of patients with DNT associated with epilepsy.

## 1. Introduction

Dysembryoplastic neuroepithelial tumors (DNT) are rare grade I neoplasms of neuroglial origin, first delineated by Daumas-Duport in 1988, that affect approximately 0.03 in 100 000 people every year in the United States [1,2,3]. It occurs mainly in children and adolescents, with an incidence peak between 10 and 14 years, and the incidence decreases sharply with age [3,4]. In most cases, affected patients present with complex partial seizures that progress to epilepsy refractory to treatment with antiepileptic drugs (AEDs) [5]. DNTs have emerged as the second most common type of low-grade epilepsy-associated neuroepithelial tumors (LEATs) after gangliogliomas, accounting for 17.8% of all LEATs in adults and 23.4% in the pediatric population [6,7]. DNTs mostly present as single lesions in the supratentorial cortex, especially in the temporal lobe [8,9]. The structural polymorphism of DNTs accounts for the heterogeneity of their imaging features, but in most cases, intracortical and well-circumscribed lesions without mass effect or perilesional edema are visible [10]. The MRI features can be classified into three subtypes that may be associated with precise surgical planning, namely type 1 MRI (cystic-like or polycystic-like), type 2 MRI (nodular-like), and type 3 MRI (dysplastic-like) [11]. Pathologically, the typical features of DNT are specific glioneuronal elements with a typical columnar structure composed of small oligodendrocytes and neurons floating within an interstitial fluid that can be divided into three subtypes based on the contents of specific components, namely the simple, complex, and nonspecific histological forms [12]. Specific DNTs, both simple and complex forms, are a cohort of homologous tumors characterized by low-grade gliomas in children: a quiescent genome with recurrent genomic alterations in the RAS-MAPK pathway, pronounced DNA methylation, suitable prognosis, but a malignant transformation in some cases. Nonspecific subsets of DNT include several recently described histomolecular entities, such as polymorphous low-grade neuroepithelial tumor of the young (PLNTY) and diffuse astrocytomas with MYB or MYBL1 alterations [13]. The pathogenesis of DNT involves alterations in constitutional and somatic fibroblast growth factor receptor 1 (FGFR1) and activation of the MAP kinase signaling pathway, which may be helpful in guiding the way to targeted therapy [14]. Given the indolent nature of DNTs, adjuvant chemotherapy or radiotherapy is often not required, and it is imperative to control the seizures by surgically removing the tumor. Factors influencing the prognosis of epilepsy surgery are primarily age at seizure onset, duration of epilepsy, tumor characteristics on MRI, histopathological findings, and extent of resection [15]. Although DNTs are generally considered to be benign, approximately 20 cases out of more than 1000 reported DNTs have shown progression, recurrence, and malignant transformation [16]. Most studies have reported favorable surgical outcomes with seizure-free rates after gross total tumoral resection of over 80% [3,15]. However, a few patients still experience seizures after surgery and have a much lower quality of life than patients without postoperative seizures [17,18,19]. A recent clinical study showed that the cumulative rate of seizure recurrence-free was 82% at one year and gradually decreased to 60% at 10 years, suggesting that the surgical outcomes of DNTs may not be favorable with longer follow-up [20]. Given its relative rarity, few studies about DNTs associated with epilepsy have been reported in the past decade. Since the sample size of these studies was limited, the long-term outcome of epileptic seizures after surgery remains unclear. To assess the long-term outcome of postoperative epileptic seizures and influential factors, we reviewed the data of 63 patients with DNTs who underwent surgical treatment at our center from January 2008 to July 2021, including seizure semiology, detailed medical history, neurological examinations, preoperative examination, surgery data, and seizure outcomes.

## 2. Materials and Methods

### 2.1. Patient Selection

In this retrospective cohort study, we reviewed all patients with pathologically confirmed DNTs from January 2008 to July 2021 in the Neurosurgery Department of Sanbo Brain Hospital of Capital Medical University, Beijing, China. Patients with DNTs surgically treated in our center were included, and patients were excluded for the following reasons: (1) absence of seizures, (2) previous history of surgery for epilepsy in other hospitals, (3) no surgical resection, (4) loss of follow-up. The patient selection process is shown in Figure 1. The present study was approved by the Ethics Committee Sanbo Brain Hospital of Capital Medical University.

### 2.2. Presurgical Assessments

Preoperative non-invasive examinations include seizure semiology, detailed medical history, neurological examinations, magnetic resonance imaging (MRI), and long-term scalp video-electroencephalogram (VEEG). Seizure types were evaluated by experienced neurosurgeons and neurologists based on the International League Against Epilepsy (ILAE) classification of epilepsies [21,22]. All patients were scanned by MRI (1.5-T, Siemens, or 3.0-T, GE). MRI sequences were T1-weighted imaging, T2-weighted imaging, and fluid-attenuated inversion recovery (FLAIR). The tumor site, volume, and MRI subtypes (Figure 2A–L) were reviewed by a neuroradiologist. Tumor dimensions were measured with a digital ruler on MRI. The measurements were based on the abnormal signals on contrast-enhanced T1WI in high-grade tumors and T2WI in low-grade tumors. The tumor volume calculation was based on the formula for the normalized volume of an ellipsoid [23]. Tumors were classified as type 1 MRI (cystic-like or polycystic-like), type 2 MRI (nodular-like), or type 3 MRI (dysplastic-like) according to the classification presented by Chassoux et al. [11]. A standard 10–20 system of electrode placement with a 64 or 128-channel system was used for long-term video-EEG monitoring. Interictal epileptiform discharges (IEDs) were termed “regional” (only involved a single lobe or adjacent lobes), “unilateral” (involved the ipsilateral hemisphere of the lesion), and “bilateral” (involved both hemispheres). For patients whose seizures could be recorded, the ictal discharge patterns were classified as the same as the IEDs. The epileptogenic zone (EZ) was identified by electrophysiologists and neurologists according to the VEEG results and semiology. 

After a routine presurgical evaluation was completed, a multidisciplinary team in our epilepsy center assessed the suitability for surgical treatment. If necessary, further non-invasive tests were performed to identify the EZ, such as magnetoencephalography (MEG) and positron emission tomography-computed tomography (PET-CT). In addition, subdural or stereo-EEG (SEEG) electrodes were implanted in patients whose epileptogenic areas were not fully aligned with the tumor or overlapped with functional areas.

### 2.3. Surgical Procedure

Surgery aimed to achieve tumoral resection as completely as possible without impairing normal function. Electrocorticography (ECoG) during the operation and other neuromonitoring facilities were performed to delineate the periphery of EZ and identify the functional areas. The extent of resection was defined according to the operative recordings, and postoperative MRI was performed within 48 h. The extent of resection was defined as “gross total resection (GTR)” with no contrast-enhancement on T1WI in high-grade tumors and hyperintensity on T2WI in low-grade tumors [23], “near-total resection (NTR)” with less than 10% of the initial tumor volume, and “subtotal resection (STR)” involved more than 10% of the tumor remnant. For tumors located in the anterior or medial temporal lobe, the standard anterior temporal lobectomy with or without medial structures was also defined as GTR. The pathological diagnosis of DNTs was confirmed by the neuropathologist. The typical features of DNT are specific glioneuronal elements with a typical columnar structure composed of small oligodendrocytes and neurons floating within an interstitial fluid (Figure 2M,N). Pathological features were divided into three subtypes confirmed by the pathologist. The simple form was composed of a specific glioneuronal element. The complex form consisted of specific glioneuronal elements combined with glial nodules and focal cortical dysplasia (FCD). The nonspecific form consisted of glial and dysplastic components without specific glioneuronal elements and multinodular structure [12,24,25].

### 2.4. Follow-Up

All patients undergoing surgical treatment were followed up postoperatively every three months in the first year and then annually on an outpatient basis by the operating neurosurgeon. MRI and scalp EEG were repeated at the first follow-up in all patients to determine whether the tumor and EZ were completely removed. All patients were routinely treated with antiepileptic drugs after surgery, and decisions were made to taper or discontinue AEDs based on postoperative seizure outcomes. Seizure outcomes were assessed according to the ILAE classification of seizure outcomes after epilepsy surgery as follows: ILAE Class 1, no seizure at all, no aura; ILAE Class 2, only auras, no other seizures; ILAE Class 3, one to three seizure days per year, with or without auras; ILAE Class 4, four seizure days per year to 50% reduction in baseline seizure days, with or without auras; ILAE Class 5, less than 50% reduction in baseline seizure days to 100% increase in baseline seizure days, with or without auras; ILAE Class 6, more than 100% increase in baseline seizure days, with or without auras. According to the ‘Rule of Three’ [26], a seizure interval of up to three times the longest inter-seizure pre-treatment interval is a seizure-free period. Patients who met this criterion were defined as seizure-free.

### 2.5. Statistical Analysis

Patients were divided into two groups according to the incidence of postoperative seizures: a seizure-free group and a non-seizure-free group. For continuous variables, means, standard deviation (SD), and ranges were presented. The continuous variables were stratified based on Kaplan–Meier analysis to identify the threshold that could affect the surgical outcomes. Categorical data were expressed as frequencies and percentages and assessed using the Pearson chi-square test or Fisher’s exact chi-square test. We used Kaplan–Meier analysis to calculate the cumulative rate of seizure recurrence-free and plotted survival curves. Subgroups were compared by log-rank (Mantel–Cox) tests. A Cox proportional hazard model was used for univariate and multivariate analyses. A *p*-value <0.05 was considered statistically significant. All statistical analyses were performed using R statistical computing software version 4.2.1 (The R Foundation). All figures and tables were created in R and Adobe Illustrator.

## 3. Results

### 3.1. Demographic Characteristics

Between January 2008 and July 2021, we reviewed 80 patients with DNTs. After excluding 17 patients, 63 patients (33 males and 30 females) were ultimately enrolled in our cohort study. The mean age at surgery was 13.54 ± 10.62 (range, 0.8–55.0) years, the mean age at seizure onset was 18.92 ± 12.44 (range, 1.0–57.0) years, and the mean duration was 4.88 ± 6.73 (range, 0.1–32.0) years.

### 3.2. Clinical Characteristics

The baseline clinical characteristics of the 63 patients grouped by seizure outcomes are presented in Table 1. We found that 18 (28.6%) patients experienced auras before the seizures. Most patients presented with only focal-onset seizures (*n* = 34, 54.0%), followed by only generalized-onset seizures (*n* = 14, 22.2%) and both seizure types (*n* = 15, 23.8%). Eight patients (12.7%) did not take AEDs preoperatively, possibly due to the short duration or low frequency of seizures, 32 (50.8%) patients were on monotherapy, and 23 (36.5%) patients received combination therapy. At the last follow-up, 37 (58.7%) patients were weaned off AEDs, 17 (27.0%) patients received monotherapy, and the remaining 9 (14.3%) patients were still on combination therapy. The mean number of AEDs after surgery (0.59 ± 0.82) was significantly lower than at baseline (1.33 ± 0.82) (*p* < 0.001).

MRI imaging showed that most tumors were located in the temporal lobe (*n* = 28, 44.4%), followed by the parietal lobe (*n* = 18, 28.6%), frontal lobe (*n* = 12, 19.0%), and occipital lobe (*n* = 3, 4.8%). MRI subtypes were categorized as type 1 MRI (*n* = 50, 79.4%), type 2 MRI (*n* = 2, 3.2%), and type 3 MRI (*n* = 11, 17.5%). The mean tumor volume was 11.26±13.02 (range, 0.25–51.81) cm^3^. Scalp EEG monitoring results were obtained for all patients. IEDs were regional in 34 patients, unilateral in 5, and bilateral in 11 patients, with nonspecific findings in 13 patients. Ictal onset rhythms were regional, unilateral, and bilateral in 18, 8, and 19 patients, respectively. During the EEG monitoring period, the seizure could not be captured in 18 (28.6%) patients. For accurate localization of epileptic foci, most patients underwent MEG (*n* = 21, 33.3%), followed by PET-CT (*n* = 4, 6.3%) and intracranial electrode implantation (*n* = 1, 1.6%). GTR was achieved in 50 patients (79.4%), while 11 patients (17.5%) underwent NTR, and the remaining 2 patients underwent STR (3.2%). Histopathological subtypes were categorized as simple form (*n* = 44, 69.8%), complex form (*n* = 16, 25.4%), and nonspecific form (*n* = 3, 4.8%). Subsequently, the neocortex surrounding the tumor was characterized by cortical disorganization in 13 (20.6%) cases, of which 6 (9.5%) had typical focal cortical dysplasia (FCD). Among patients with FCD (*n* = 6), five were classified as FCD 3b and one with FCD 2a according to the FCD classification criteria published by ILAE in 2011 [27].

### 3.3. Surgical Complications

Of 63 patients, 9 (14.3%) had surgery-related complications, comprising temporary (*n* = 6, 9.5%) and permanent (*n* = 3, 4.8%) complications. Temporary complications included contralateral limb muscle weakness that recovered at post-hospital discharge (*n* = 4, 6.3%), transient paresthesia (*n* = 1, 1.6%), and intracranial infection requiring debridement (*n* = 1, 1.6%). Permanent complications included fine motor disability (*n* = 1, 1.2%) and hypomnesia associated with temporal lobe procedures (*n* = 2, 2.4%). No deaths occurred postoperatively.

### 3.4. Follow-Up and Outcomes

All patients were followed up for at least one year, with a mean follow-up duration of 7.01 ± 3.46 years (1.62–13.22). At the last follow-up, the 63 patients were classified as ILAE 1 (*n* = 43, 68.3%), ILAE 2 (*n* = 1, 1.2%), ILAE 3 (*n* = 12, 19.0%), ILAE 4 (*n* = 2, 3.2%), ILAE 5 (*n* = 3, 4.8%) and ILAE 6 (*n* = 2, 3.2%). Six patients experienced <3 seizures on the same or different days after surgery and were seizure-free at all other times. According to the ‘Rule of Three’ [26], we included six patients in the group who achieved seizure freedom, of whom three patients had seizures in the first postoperative year (seizure frequency: two patients had a seizure once, and one had seizures twice on different days); one patient had a seizure once due to tumor recurrence in the fourth year and achieved seizure-free after reoperation; two patients had seizures twice (one in the first and third year, another in the third and eighth year, respectively). Overall, 77.8% (49/63) of patients were seizure-free. Further details are available in Figure 3.

### 3.5. Univariable Survival Analysis

Kaplan–Meier analysis showed that the cumulative rate of seizure recurrence-free was 82.5% (95% confidence interval (CI) 73.7–92.5%), 79.0% (95% CI 69.5–89.9%), and 76.5% (95% CI 66.2–88.3%) at 2, 5, and 10 years, respectively (Table A1). The mean time for seizure recurrence-free was 6.892 ± 0.501 years (95% CI 5.91–7.87). The probability fluctuated in the first few years after surgery and gradually stabilized in subsequent years (Figure 4A).

The univariable and pairwise comparisons were analyzed by the log-rank (Mantel–Cox) test shown in Table A2. There were statistically significant differences among age at seizure onset (*p* = 0.007), duration of seizures (*p* = 0.002), scalp interictal VEEG (*p* = 0.003), MRI subtype (*p* = 0.003), and type of surgery (*p* ˂ 0.001). Figure 3 shows the survival curves of statistically significant variables. Overall, we found that age at seizure onset >4 years, regional IEDs, and GTR were associated with favorable seizure outcomes. In contrast, the duration of seizures >4 years and MRI type 3 correlated with a poor prognosis. No association was found between other variables and seizure outcomes.

### 3.6. Multivariable Cox Regression Analysis

First, a univariable Cox regression analysis was performed (Table A3). Parameters with a *p*-value < 0.05 during univariate analysis were incorporated into the multivariate analysis. The final multivariable Cox regression model consisted of age at onset, duration, IEDs, MRI subtype, and surgical type.

The final multivariable Cox regression model with subgroup analysis is presented in Table 2. Interestingly, we found a significant association between GTR and postoperative seizure freedom (hazard ratio (HR) 6.38, 95%CI 1.18–34.37, *p* = 0.031). The same association was found between the duration of seizures of less than 4 years and postoperative seizure freedom (HR 4.32, 95%CI 1.31–14.2, *p* = 0.016). Regional IEDs exhibited a more significant correlation with postoperative seizure freedom than bilateral (HR 6.38, 95%CI 1.03–39.64, *p* = 0.047) IEDs or unilateral (HR 10.09, 95%CI 1.41–72.09, *p* = 0.021) IEDs in the univariable Cox regression model. In addition, unfavorable postoperative seizure freedom was associated with age at seizure onset of more than 4 years (HR 0.25, 95%CI 0.08–0.74, *p* = 0.012) and MRI type 3 (HR 4.26, 95%CI 1.47–12.32, *p* = 0.007) in the univariable Cox regression model.

## 4. Discussion

Herein, we substantiated the benefit of surgical treatment for intractable epilepsy in DNT patients. At the last follow-up, 49 of 63 patients (77.8%) were seizure-free, similar to past studies, which reported a rate of 68-83% [6,7,8]. At a mean follow-up of 84.46 ± 41.54 (range, 19.4–158.6) months, the seizure-free rate remained essentially stable at approximately 80% based on a year-to-year analysis of seizure outcomes. Moreover, the cumulative rate of seizure recurrence-free postoperatively was 82.5%, 79.0%, and 76.5% at 2, 5, and 10 years, respectively, which was higher than reported by Yang et al. (73.0%, 70.0%, and 60.0% at 2, 5, and 10 years) [20]. In the present study, we comprehensively explored factors associated with postoperative outcomes. After the univariate and multivariate analyses, GTR and a short seizure duration were the most important factors associated with postoperative seizure freedom. Other statistically significant variables associated with poor prognosis included younger seizure onset age, bilateral IEDs, and MRI type 3.

In this long-term cohort study, survival analysis indicated that GTR (79.4%) was the most important long-term predictor, consistent with the literature [11,28,29,30]. However, the extent of surgical resection has long been subject to debate, with reports of complete resection rates ranging from 42.3% to 100% [28,31,32]. Daumas-Duport et al. and Morris et al. reported that incomplete resection of tumors could also lead to a favorable prognosis in the initial case series [1,33], whereas Chassoux et al. highlighted the importance of extended resection of tumors to the surrounding epileptogenic cortex [11]. Yang et al. believed that the discovery of satellite lesions around the tumor, which may be a tumor of independent origin, together with extended surgical resection, is more favorable for the prognosis [20]. In addition, tumors of the medial temporal lobe should be removed together with the medial structure after detailed evaluation in carefully selected pediatric patients [34,35]. Targeting tumors in the motor function region may be daunting due to the potential for permanent complications [36]. In the present study, the total tumor removal rate in the functional region was 33.3% (2/6), and two patients were postoperatively seizure-free, including one with transient muscle weakness in the upper limb and one with long-term fine motor disability. Therefore, individualized assessment of the patient’s risk-benefit ratio is critical prior to surgery. In addition, MRI and VEEG should be conducted as early as possible in postoperative resected patients to predict the likelihood of recurrence and take appropriate measures, such as medication modification or reoperation.

A study pointed out that the probability of seizure freedom was higher in tumors in the temporal lobe [29], while another proposed that extratemporal tumors were associated with better seizure outcomes [18]. Although DNTs lesions were predominantly found in temporal lobes (46%), we found no significant association with prognosis between temporal and extratemporal locations. Cortical dysplasia frequently occurs with DNT, predominantly in children, representing further extension into the peritumoral cortex, which might be the structural basis of seizure recurrence [9,15,34,37]. Although CD may contribute to overall epileptogenicity, many studies did not find an association between CD and seizure outcome [29,32,37,38]. In our study, cortical disorganization in tissue adjacent to the tumor was noted in 13 (20.6%) patients, of which 6 cases had typical FCD features. Five patients (83.3%) with FCD achieved seizure freedom, suggesting that cortical dysplasia was not associated with a worse prognosis, which may be attributed to the routine use of intraoperative electrocorticography (ECoG) as an adjunct to guide the resection of extralesional tissue in our center. Therefore, it is important to delineate the extent of extratumoral EZ under ECoG guidance. In addition, a recent study suggested that applying high-frequency oscillations (HFOs) in ECoG instead of epileptic spikes may have better outcomes [39]. Given the lack of detailed data on HFOs in our patients, more reports may be needed to validate this hypothesis.

In the present study, we found that older seizure onset age was a positive factor only in the univariate Cox regression analysis. In contrast, Cai et al. recently reported that older age of onset was associated with worse seizure outcomes [18]. Meanwhile, some studies suggested that the age at seizure onset was irrelevant to seizure outcome [29,32,40]. Although the exact reason remains unclear, we hypothesize that seizure onset at a younger age may lead to more severe damage to the peritumoral cortex since the prime period of brain development occurs in the first few years after birth. In addition, Nolan et al. and Isler et al. demonstrated a significant benefit of shorter duration and younger age at surgery [28,41]. In our cohort study, only a duration of seizure onset >4 years was associated with poor prognosis, consistent with the literature [11,18,34,35]. In the long term, tumors and recurrent seizures can cause damage to the peritumoral cortex, leading to cognitive impairment and seriously affecting the quality of life [31,42]. Accordingly, we encourage early surgical intervention for patients with DNTs, especially children. 

Lee et al. and Mittal et al. suggested that an unfavorable seizure outcome was associated with bilateral epileptiform discharges [43,44]. We only found an association with bilateral IEDs in multivariable analysis. Bilateral epileptiform discharges may indicate secondary epilepsy foci in other brain regions that result from recurrent seizures, leading to a poor prognosis [44]. However, the results of the bilateral ictal rhythms were inconclusive. We hypothesize that for patients with low seizure frequency where it was challenging to capture the seizure during monitoring, direct surgery was indicated, guided by the MRI-based localization of the lesion and the results of IEDs. The lack of sufficient data on ictal rhythms led to inconclusive results. In addition, patients require further evaluations, including MEG, PET-CT, and SEEG, to identify the EZ that scalp EEG cannot accurately localize, especially in deep brain tumors.

Chassoux et al. studied 33 patients who underwent SEEG and found that the epileptogenic zone was colocalized with the tumor in type 1 MRI, included the peritumoral cortex in type 2 MRI, and involved extensive regions in type 3 MRI [11]. This may explain the poor prognosis of patients with type 3 MRI after GTR in the present study. Isler et al. found that the seizure-free rate was 80.0% (8/10) in type 1 MRI, 71.4% (5/7) in type 2 MRI, and 100% (4/4) in type 3 MRI, separately [41]. It is widely thought that in some cases, patients can be treated with extended resection under intraoperative ECoG monitoring to improve the prognosis of epilepsy in patients with MRI types 2 and 3. In addition, Chassoux et al. found that type 1 MRI always corresponded to simple or complex DNTs, while type 2 MRI and type 3 MRI corresponded to nonspecific forms, analogous to results reported by Cai et al. and Isler et al. [11,18,41,45]. In our study, only consistent findings were observed for type 1 MRI, and correlations for the other subtypes were not available. It is worth noting that there was no statistical difference between pathological subtypes and prognosis, which may be related to the fact that our pathological subtypes were mainly simple forms (73%), which differed significantly from previous studies where complex forms 2–3 times occurred more than simple forms [4,46]. Accordingly, no definitive conclusions can be drawn, and more clinical reports are needed to validate them. 

Neurocognitive dysfunction is one of the most common postoperative complications of brain tumors, which severely impacts patients’ quality of life. For various reasons, our data on preoperative and postoperative neurocognitive outcomes are not available. The occurrence of cognitive impairment is related to the direct effect of tumors, seizures, and/or resection under general anesthesia. It is currently believed that abnormal expression of cytokines, including inflammatory factors [47,48,49,50], reactive oxygen species [51], and high mobility group protein B1 (HMGB1) [52], are the molecular mechanism of neurocognitive dysfunction. Tumors in different regions can cause different cognitive impairments, such as speech and executive disorders in language-functional areas, naming disorders in the temporal and frontal lobes, aprosexia in the frontal lobes, naming disorders, and mild memory impairment in the insular lobes [53]. Postoperative cognitive function is significantly related to preoperative cognitive function, which is associated with younger age of onset and longer duration of epilepsy [54]. DNTs can provide a unique cognitive model to study the effects of early vs. late onset, suggesting that early childhood seizures and treatments may impair core cognitive development and lead to severe cognitive deficit patterns in adulthood [55]. Removal of brain tissue and injury to the brain or nerves caused by improper surgery can also lead to cognitive changes. A study involving 59 patients with WHO grade I-III diffuse gliomas found that 17% of patients had cognitive improvement and 42% of patients experienced postoperative cognitive decline, 17% of whom experienced a cognitive decline in multiple domains, most commonly affecting attention (17%) and information processing speed (15%) [56]. Faramand et al. reported a study of 150 LEATs that showed that postoperative full-scale intelligence quotient (FSIQ) improved in 61% of children, decreased in 36.5%, and remained unchanged in 2.5% [57]. Therefore, earlier surgery can improve postoperative FSIQ scores in children, especially young children, and cognitive function after epilepsy surgery is expected to be improved. Due to the abstract nature of cognitive function, a series of scales have been designed clinically to better assess neurocognitive dysfunction [58,59,60]. However, how to choose the appropriate approach for different patients is still inconclusive. The treatment of neurocognitive dysfunction includes drug therapy and nondrug therapy [61], which is still in the preliminary stage of research and lacks clinical guidelines.

We recognize several limitations to our study. First, the retrospective nature of the present study brought inherent limitations, such as poor control factors and potential biases. Second, other factors related to the prognosis of epilepsy-associated tumors were not considered, such as BRAF V600E mutation rate and CD34 positivity rate. Moreover, the continuous variables were stratified based on Kaplan–Meier analysis, and the results may be variant with clinical stratification. Finally, given the limited sample size and the fact that the final Cox risk proportional model did not incorporate all the variables of interest to us, the results of multivariable Cox regression analysis may be biased to a certain extent.

## 5. Conclusions

This single-center retrospective cohort study provided compelling evidence that most DNT patients present epilepsy as their clinical symptom and can be treated with surgical resection. Gross total removal of the tumors and shorter duration are associated with a better seizure outcome, while younger onset, bilateral IEDs, and type 3 MRI are associated with poor prognosis. These characteristics are helpful for clinicians in predicting the prognosis of epilepsy associated with DNTs.

## Figures and Tables

**Figure 1 brainsci-13-00024-f001:**
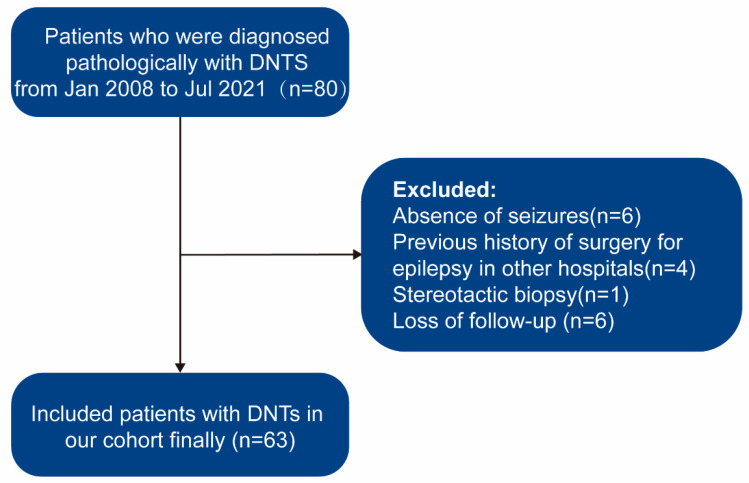
Flowchart describing the procedures and exclusion and inclusion criteria of this study. DNTs, dysembryoplastic neuroepithelial tumors.

**Figure 2 brainsci-13-00024-f002:**
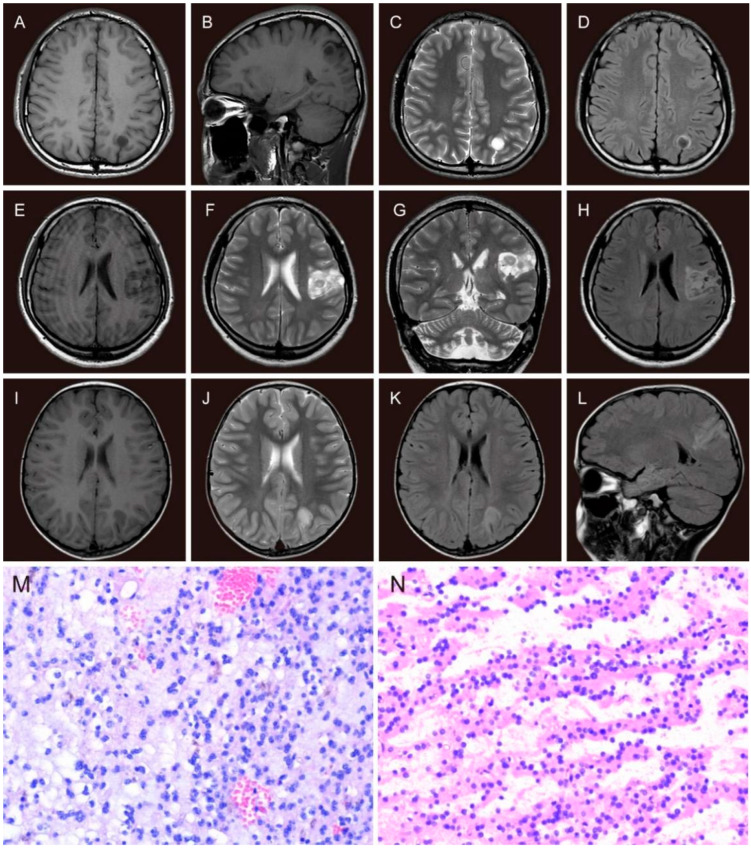
The MRI features and histopathological features of DNTs. The MRI features of type 1 (**A**–**D**): axial and sagittal slices, single cystic-like hypointensity on T1WI (**A**,**B**) and hyperintensity on T2WI (**C**), ring hyperintensity around the isointensity tumor with a clear gray-white matter boundary on FLAIR image (**D**); the MRI features of type 2 (**E**–**H**): axial and coronal slices, hypointensity on T1WI (**E**), hyperintensity on T2WI (**F**,**G**) and FLAIR image (**H**), all with a nodular-like iso-hypointensity; the MRI features of type 3 (dysplastic-like): axial and sagittal slices (**I**–**L**), slightly blurring of the gray-white matter demarcation with hypointense signal on axial T1WI (**I**), hyperintense signal on T2WI (**J**) and FLAIR images (**K**,**L**). Histopathological features of DNT (**M**,**N**): specific glioneuronal elements with a typical columnar structure composed of small oligodendrocytes and neurons floating within an interstitial fluid (H&E, ×200); (**N**): tumor cells were “striped” in loose areas of tumor tissue (H&E, ×200).

**Figure 3 brainsci-13-00024-f003:**
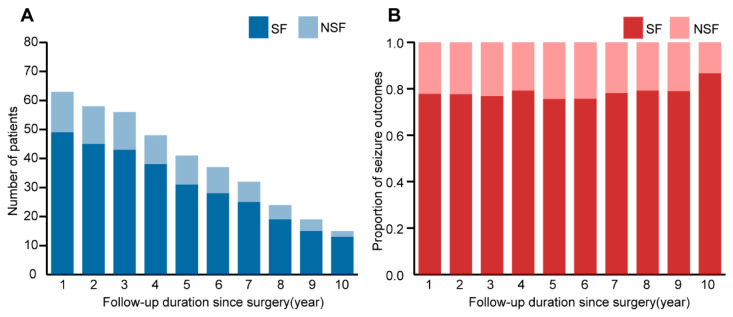
(**A**): Number of patients by seizure outcomes according to follow-up duration since surgery; (**B**): year-to-year analysis of seizure outcomes, the seizure-free rate remained basically stable.

**Figure 4 brainsci-13-00024-f004:**
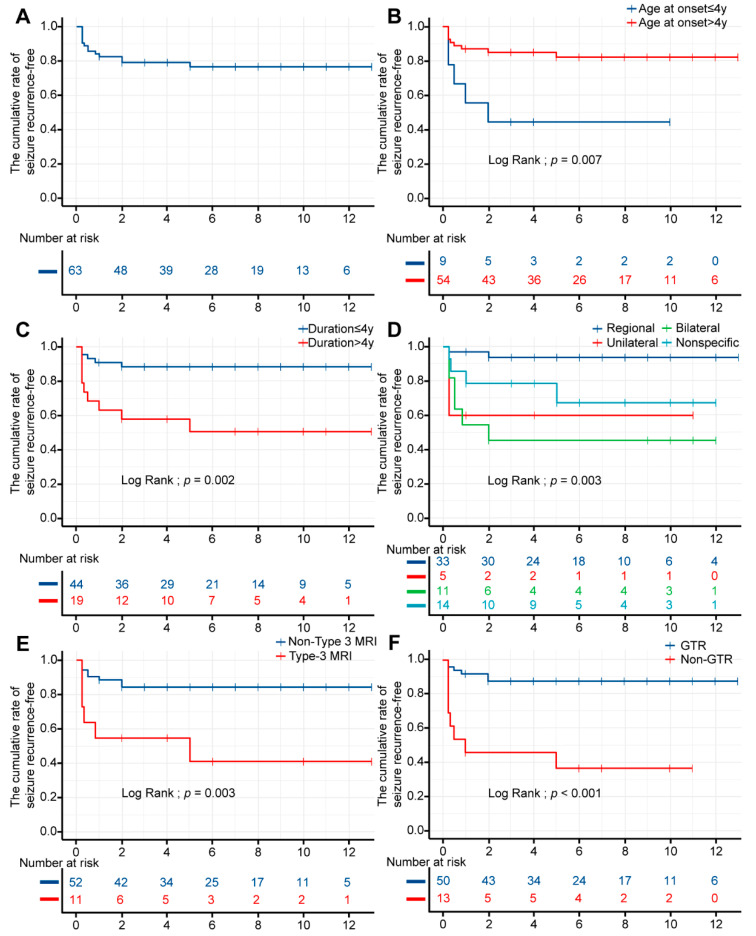
Kaplan–Meier curves for seizure outcomes. The X-axis represents time in years, and the Y-axis represents the proportion. (**A**): The rate fluctuated in the first few years after surgery and gradually stabilized in subsequent years. (**B**): Analysis by age at onset; (**C**): analysis by duration; (**D**): analysis by IEDs; (**E**): analysis by MRI subtype; (**F**): analysis by surgical types. Addition: because of the limited sizes of MRI type 2 and NTR, MRI type 1 and 2 were classified as Non-Type 3, STR and NTR were classified as Non-GTR in this Kaplan–Meier curve.

**Table 1 brainsci-13-00024-t001:** Demographic and clinic characteristics of patients and the relationship with seizure outcomes.

Characteristics	SF (*n* = 49)	NSF (*n* = 14)	*p*-Value
Sex			
Male	28 (57.1%)	5 (35.7%)	0.266
Female	21 (42.9%)	9 (64.3%)	
Age at seizure onset			
≤4 years	4 (8.2%)	5 (35.7%)	0.030 *
>4 years	45 (91.8%)	9 (64.3%)	
Duration of seizures			
≤4 years	39 (79.6%)	5 (35.7%)	0.005 *
>4 years	10 (20.4%)	9 (64.3%)	
Age at surgery			
≤16 years	27 (55.1%)	6 (42.9%)	0.613
>16 years	22 (44.9%)	8 (57.1%)	
Aura			
Yes	35 (71.4%)	10 (71.4%)	1.000
No	14 (28.6%)	4 (28.6%)	
Seizure frequency			
Daily	10 (20.4%)	1 (7.1%)	0.654
Weekly	15 (30.6%)	4 (28.6%)	
Monthly	17 (34.7%)	7 (50.0%)	
Yearly	7 (14.3%)	2 (14.3%)	
Seizure types			
Focal only	27 (55.1%)	7 (50.0%)	0.848
Generalized only	10 (20.4%)	4 (28.6%)	
Both	12 (24.5%)	3 (21.4%)	
IEDs			
Regional	31 (63.3%)	2 (14.3%)	0.002 *
Unilateral	3 (6.1%)	2 (14.3%)	
Bilateral	5 (10.2%)	6 (42.9%)	
Nonspecific	10 (20.4%)	4 (28.6%)	
Ictal onset rhythms			
Regional	16 (32.7%)	2 (14.3%)	0.523
Unilateral	6 (12.2%)	2 (14.3%)	
Bilateral	13 (26.5%)	6 (42.9%)	
Not captured	14 (28.6%)	4 (28.6%)	
Laterality of tumor in preoperative MRI			
Left	26 (53.1%)	8 (57.1%)	1.000
Right	23 (46.9%)	6 (42.9%)	
Site of lesion			
Temporal	21 (42.9%)	8 (57.1%)	0.521
Extratemporal	28 (57.1%)	6 (42.9%)	
MRI subtype			
Type 1	42 (85.7%)	8 (57.1%)	0.020 *
Type 2	2 (4.1%)	0 (0%)	
Type 3	5 (10.2%)	6 (42.9%)	
Surgical type ^1^			
GTR	44 (89.8%)	6 (42.9%)	<0.001 *
NTR	5 (10.2%)	6 (42.9%)	
STR	0 (0%)	2 (14.3%)	
Histopathological types ^2^			
Simple form	36 (73.5%)	10 (71.4%)	0.146
Complex form	12 (24.5%)	2 (14.3%)	
Nonspecific form	1 (2.0%)	2 (14.3%)	
Complication			
No	42 (85.7%)	12 (85.7%)	0.822
Temporary	5 (10.2%)	1 (7.1%)	
Permanent	2 (4.1%)	1 (7.1%)	

Abbreviations: MRI, magnetic resonance imaging; IED, interictal epileptic discharge; GTR, gross total resection; NTR, near-total resection STR, subtotal resection. ^1^ Based on surgical records and postoperative neuroimaging. ^2^ Based on histopathological examination. * *p* < 0.05.

**Table 2 brainsci-13-00024-t002:** Relative variables for seizure recurrence estimated with a Cox proportional hazards model on univariable and multivariable Cox regression analysis.

Variables	Univariable Analysis			Multivariable Analysis		
	HR	*p*-Value	95%CI	HR	*p*-Value	95%CI
Age at seizure onset						
≤4 years	1.00					
>4 years	0.25	0.012 *	0.08–0.74	0.94	0.939	0.20–4.43
Duration of seizures						
≤4 years	1.00					
>4 years	4.87	0.005 *	1.63–14.56	4.32	0.016 *	1.31–14.2
IEDs						
Regional	1.00					
Unilateral	10.09	0.021 *	1.41–72.09	4.91	0.130	0.63–38.47
Bilateral	12.09	0.002 *	2.43–60.09	6.38	0.047 *	1.03–39.64
Nonspecific	5.33	0.054	0.97–29.13	1.26	0.817	0.18–8.91
MRI subtype						
Type 1	1.00					
Type 2	0.00	0.999	0.00–Inf	0.00	0.999	0.00–Inf
Type 3	4.26	0.007 *	1.47–12.32	1.31	0.705	0.33–5.25
Surgical type						
GTR	1.00					
Non-GTR	7.86	<0.001 *	2.70–22.95	6.38	0.031 *	1.18–34.37

Abbreviations: MRI, magnetic resonance imaging; IEDs, interictal epileptic discharges; GTR, gross total resection; Non-GTR, near-total resection and subtotal resection; HR, hazard ratio; CI, confidence intervals. * *p* < 0.05.

## Data Availability

The data presented in this study are available in Results and Appendix A.

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
