# Peer review of "Long-Term Seizure Outcomes and Predictors in Patients with Dysembryoplastic Neuroepithelial Tumors Associated with Epilepsy"

_brainsci, 2022, doi:10.3390/brainsci13010024_

Round 1

Reviewer 1 Report

This article is overall well written. Authors presented convincing results. 

however, some minor point should be addressed.

Minor

Intro:

"that affect approximately 0.03 35 in 100 000 people every year[" Is it worldwide ? 

Please could you clearly state the hypothesis and objectives of this study ? 

Do you have the reason why patients dropped the study ? 

I think the Introduction could be improved with a more detailed description of the disease and some risk factor (like mutation) if they are known. .

Last but not least, I don't see any mention of the ethic protocol. 

Reviewer 2 Report

The authors propose a single-center retrospective analysis of long-term seizure outcomes and predictors in patients with epileptogenic DNTs. Overall, they enrolled a small cohort of patients (63) but they accurately analyzed neuro-oncological and epileptogenic features and conducted a rigorous statistical analysis providing interesting results. The paper is brilliantly written, and the figures and tables are clear and of good quality. Although the topic is well known in the literature, the authors propose statistically valid results that may be useful to the neurosurgical community.

However, in my opinion, there would be some minor revisions to be performed:

- the extent of removal of the lesion should be better defined; authors should include an adequate reference on how they calculated EOR (for example Sanai et al.)

- In addition to the epileptic outcome, the authors should also analyze the long-term neurocognitive outcomes as they are possibly impaired in patients undergoing surgery. If data are not available, it would be useful to include a chapter on this in the discussion.

Reviewer 3 Report

The authors' hypothesis was to determine the predictors and long-term outcomes of patients with seizures following surgery for DNTs. However, the authors' results didn't support that hypothesis, as the results were not statistically significant. This paper is more suitable as a technical descriptive paper for the surgical technique as it doesn't add any value to the short-term or long-term seizure outcomes after surgical resection. 

Round 2

Reviewer 2 Report

The article is now suitable fo publication